# Evaluation of Baseline Characteristics and Prognostic Factors in Multisystemic Inflammatory Syndrome in Children: Is It Possible to Foresee the Prognosis in the First Step?

**DOI:** 10.3390/jcm11154615

**Published:** 2022-08-08

**Authors:** Benhur Sirvan Cetin, Ayşenur Paç Kısaarslan, Sedanur Tekin, Merve Basol Goksuluk, Ali Baykan, Başak Nur Akyıldız, Yılmaz Seçilmiş, Hakan Poyrazoglu

**Affiliations:** 1Division of Pediatric Infectious Diseases, Department of Pediatrics, Faculty of Medicine, Erciyes University, 38039 Kayseri, Turkey; 2Division of Pediatric Rheumatology, Department of Pediatrics, Faculty of Medicine, Erciyes University, 38039 Kayseri, Turkey; 3Department of Pediatrics, Faculty of Medicine, Erciyes University, 38039 Kayseri, Turkey; 4Department of Biostatistics, Faculty of Medicine, Erciyes University, 38039 Kayseri, Turkey; 5Division of Pediatric Cardiology, Department of Pediatrics, Faculty of Medicine, Erciyes University, 38039 Kayseri, Turkey; 6Division of Pediatric Intensive Care Unit, Department of Pediatrics, Faculty of Medicine, Erciyes University, 38039 Kayseri, Turkey; 7Division of Pediatric Emergency, Department of Pediatrics, Faculty of Medicine, Erciyes University, 38039 Kayseri, Turkey

**Keywords:** COVID-19, multisystem inflammatory syndrome in children (MIS-C), risk classification, prognosis

## Abstract

Background: Two years after the first cases, critical gaps remain in identifying prognostic factors in multisystem inflammatory syndrome in children (MIS-C). Methods: This retrospective study included 99 patients with MIS-C hospitalized between August 2020 and March 2022 in a pediatric tertiary center. The patients were divided into two groups according to clinical severity (low- and high-risk). Prognostic values of baseline clinical and laboratory characteristics were evaluated with advanced statistical analysis, including machine learning. Results: Sixty-three patients were male, and the median age was 83 (3–205) months. Fifty-nine patients (59.6%) were low-risk cases. Patients aged six years and over tended to be at higher risk. Involvement of aortic or tricuspid valve or >1 valve was more frequent in the high-risk group. Mortality in previously healthy children was 3.2%. Intensive care unit admission and mortality rate in the high-risk group were 37.5% and 7.5%, respectively. At admission, high-risk patients were more likely to have reduced lymphocyte count and total protein level and increased brain natriuretic peptide (BNP), ferritin, D-dimer, and troponin concentrations. The multiple logistic regression model showed that BNP, total protein, and troponin were associated with higher risk. When the laboratory parameters were used together, BNP, total protein, ferritin, and D-dimer provided the highest contribution to the discrimination of the risk groups (100%, 89.6%, 85.6%, and 55.8%, respectively). Conclusions: Our study widely evaluates and points to some clinical and laboratory parameters that, at admission, may indicate a more severe course. Modeling studies with larger sample groups are strongly needed.

## 1. Introduction

Multisystem inflammatory syndrome in children (MIS-C) is an immune dysregulation after SARS-CoV-2 infection [1]. In all treatment guidelines, intravenous immunoglobulin (IVIG) and steroids are recommended primarily for MIS-C treatment [2,3,4,5,6]. However, due to the high cost of treatment and difficulties in accessing IVIG from time to time, the search for optimal treatment continues. Two years after the first cases, critical gaps remain in identifying prognostic factors in multisystem inflammatory syndrome in children. Clinical studies for early differentiation of cases with low risk of complications and good clinical course will help shape more cost-effective MIS-C treatment protocols. This study assesses the initial clinical and basic laboratory features associated with severe disease and complications in MIS-C.

## 2. Materials and Methods

This study is a retrospective analysis of patients aged 1 month to 18 years who were diagnosed with MIS-C between August 2020 and March 2022. We considered patients to have confirmed MIS-C if they fulfilled MIS-C clinical criteria of WHO or CDC, and SARS-CoV-2 or antibodies to SARS-CoV-2 were detected at any point [2,3]. Patients were treated in accordance with Erciyes University MIS-C Guideline [7], which was based on the Clinical Guidance of the American College of Rheumatology (ACR) [5].

Cases were divided into two groups for risk classification. The criteria were determined according to the clinical findings observed during admission and hospitalization. The feature that we considered as a risk here was defined as the risk of a more severe course of the disease. Patients who had an uncontrolled chronic disease or had a severe clinical condition during hospitalization, needed high-dose steroids, oxygen, or inotropic support, or had cardiac involvement or had a hospital stay longer than 10 days, were defined as the high-risk group (Figure 1). Patients who did not meet these characteristics were defined as the low-risk group.

A severe clinical condition was defined as the presence of hypotension, multiple organ failure (MOF), central nervous system involvement, or intensive care unit admission. MOF is defined as the physiological dysfunction of two or more organ systems where homeostasis cannot be maintained without intervention. Patients with altered consciousness or other neurological symptoms and findings were considered central nervous system (CNS) involvement. Isolated headache not accompanied by any other neurological finding was not considered CNS involvement. Hypotension is defined as less than the 5th percentile of age and gender-specific blood pressure levels. Echocardiography findings define cardiac involvement. Left ventricular dysfunction was defined as decreased EF (<55%) or FS (<28%). Coronary artery abnormalities were defined according to the AHA guidelines [8]. During the study period, our center analyzed troponin levels with different laboratory kits, with troponin levels categorized into two groups: normal or <10-fold of the upper limit, and ≥10-fold of the upper limit. For SARS-CoV-2 serology, rapid antibody test kits (Colloidal Gold COVID-19 IgM/IgG Antibody Rapid Test, HotGen^©^ Biotech Co., Beijing, China) were used.

### Statistical Analyses

The IBM SPSS Statistics for Windows version 26.0 and R language environment (version 4.0.5, URL: https://cran.r-project.org (accessed on 10 April 2022) were used for statistical analysis. The variables were investigated using visual (histogram, probability plots) and analytic methods (Kolmogorov–Smirnov, Shapiro–Wilk tests) to determine whether they were normally distributed. Descriptive analysis was presented using mean ± standard deviation (SD) or median (minimum (min.)–maximum (max.) values) for continuous variables and frequency (percentage) for categorical variables. Categorical data were statistically analyzed by Fisher’s exact or chi-square tests, as appropriate. Continuous data were analyzed by Student’s t-test or Mann–Whitney U test, whichever was appropriate. Multiple logistic regression was used to determine which factors may affect the risk status. To determine the classification performance of each biomarker, separately, the area under the ROC curve was used. The cut-off value of each biomarker was obtained using the Youden index. *p* values < 0.05 were considered statistically significant. In addition, the classification performance of combined tests and their contributions were evaluated with machine learning algorithms (decision tree, support vector machine, logistic regression analysis, naïve Bayes, and linear discriminant analysis) using the “caret” package in R. Our aim was to explore significant predictors of risk groups. In the first step, we performed a simple logistic regression model for each predictor separately. The predictors with *p* values below 0.10 were selected as possible risk factors and the set of all possible predictors was included in the multiple logistic regression model. In this model, we adjusted the effect of each predictor by the remaining set, with the statistical significance level set as *p* < 0.05 in the final model. To conclude, *p* < 0.10 was used to find the list of predictors via the crude model, while *p* < 0.05 was used to find the significant predictors in the multiple models (i.e., covariate-adjusted model).

## 3. Results

### 3.1. Distribution of Patients in the Study Period

During the study period, 105 children were diagnosed with MIS-C (99 confirmed, 6 probable) at our center, with only confirmed patients included in the study. The median age of patients was 83 (min.-max. values 3–205) months and 63 (63.6%) of the patients were male. The first case was seen in August 2020 at our center. The first peak in the number of cases was seen in November 2020. There was a rapid decrease in cases in the following months. In the fourth quarter (Q4) of 2021, the number of cases climbed again, with the second peak seen in October 2021 (Figure 2). The total number of cases was 34 and 25, respectively, in the Q4 of 2020 and 2021, when the highest number of cases were seen. While the rate of high-risk cases was 58.8% (20/34) in the Q4 of 2020, this rate decreased significantly to 24% (6/25) in the Q4 of 2021 (*p* < 0.05). In terms of age and gender distribution of the cases, there was no significant difference between the overall and fourth quarters of 2020 and 2021.

### 3.2. Clinical Characteristics and COVID-19 Related Data of Patients

Detailed clinical features and data related to COVID-19 of the patients are presented in Table 1. The high-risk group comprised 40.4% of the cases (40/99). The median age was not different between risk groups. However, 75% (30/40) of the patients in the high-risk group were ≥72 months old, with this rate lower (54.2%, 32/59) in the low-risk group (*p* = 0.03). The male gender was similar in both groups (64.4% vs. 62.5%). Only 10 cases (10.1%) had an underlying disease. The main ones were autoinflammatory diseases, asthma, metabolic disease, and malignancy. Concomitant Kawasaki Disease (KD) findings in MIS-C patients, in order of frequency, were conjunctivitis (69.7%), rash (64.6%), oral mucosal changes (48.5%), peripheral extremity changes (30.3%), and cervical lymphadenopathy (13.1%). The frequency of these findings was not different between risk groups. When we look at the number of KD criteria met at the time of diagnosis, except fever, 2 or 3 criteria were positive in 52.5% of the cases (52/99). In 15 cases (15.2%), there were ≥4 criteria positivity for KD at the time of diagnosis. There was no difference in the incidence of KD findings between risk groups. When other accompanying symptoms and findings were examined, the most common symptoms were abdominal pain (67.8%), nausea–vomiting (64.6%), diarrhea (47.5%), and headache (30.6%). There was no difference in the incidence of these findings between the risk groups. When the history of COVID-19 was questioned, 25.3% (25/99) of the cases had a documented SARS-CoV-2 infection, while 43.4% (43/99) had a history of close contact without evidence of infection. In 31.3% (31/99) of the cases, there was neither a history of COVID-19 nor a close contact. Of those with a history of infection, 92% (23/25) had a mild infection. In cases with a history of COVID-19 or close contact, it was observed that MIS-C developed in a median of 4 weeks (min.-max. 2–16 weeks) after exposure to the virus. The median time between the appearance of MIS-C findings and admission to the hospital was 5 days (1–10 days). In cases of close contact or illness, there was no difference between the risk groups in the time elapsed between exposure to SARS-CoV-2 and the development of MIS-C and between the onset of symptoms and admission to the hospital.

The incidence of abnormal findings on chest X-rays at admission was significantly higher in the high-risk group (47.5% (19/40) vs. 8.5% (5/59), *p* < 0.001). At admission, 13.1% (13/99) of the cases were hypotensive and 15.2% (15/99) required respiratory support. The rate of hospitalization in the intensive care unit was 15.2% (15/99). MOF rate was 9% (9/99). The median length of hospital stay was 7 days (min.-max. 3–24 days). Twenty-three cases (23.2%) had a hospital stay of 10 days or longer. The most common clinical conditions that caused the cases to be included in the high-risk group were prolonged hospitalization (23/99), respiratory failure (15/99), and hypotension (13/99), respectively. The mortality rate was 3% (3/99) in general, 1.1% (1/89) in previously healthy children, and 20% (2/10) in children with underlying diseases (*p* = 0.03).

### 3.3. Laboratory and Cardiac Findings of Patients

Six cases with chronic diseases (two malignancies, two autoinflammatory, one chronic kidney disease, and one metabolic disease) had baseline abnormalities before MIS-C, so they were excluded during the analysis of laboratory values. Laboratory test results of the patients at admission are presented in Table 2. In our study, all laboratory values were compared between low- and high-risk groups, and selected ones are shown in Figure 3. In low-risk and high-risk groups, respectively, the median values of absolute lymphocyte count (ALC) were 1020 cell/mm^3^ and 775 cell/mm^3^, brain natriuretic peptide (BNP) were 2694 pg/mL and 11,289 pg/mL, ferritin were 437 ng/mL and 894 ng/mL, D-dimer were 3530 and 6305 ng/mL, and total protein were 5.94 g/dL and 5.39 g/dL. Higher BNP, ferritin, and D-dimer levels and lower ALC and total protein levels were detected in high-risk patients (*p* < 0.001, *p* < 0.001, *p* < 0.005, *p* < 0.05, and *p* < 0.001, respectively). White blood cell (WBC), platelet, C-reactive protein (CRP), procalcitonin (PCT), albumin, and other laboratory values were not statistically different between the risk groups.

For all patients, echocardiography results were evaluated in Table 3. Valvulitis was present in 72.7% (72/99) of patients. The heart valves involved, in order of frequency, were mitral (71.7%), tricuspid (25.3%), aortic (10.1%), and pulmonary (3%) valves. Valvulitis in tricuspid and aortic valves were significantly more common in high-risk patients (*p* = 0.001 and *p* = 0.007, respectively). When the number of valves involved was evaluated, it was observed that two or more valves involved were more frequent in the high-risk group (*p* = 0.004). While 27.3% (27/99) of the cases did not have valvular regurgitation, the first, second, and third-degree valve insufficiency rates were 65.7%, 6.1%, and 1%, respectively. The second and third-degree insufficiency rates were higher in the high-risk group than in the low-risk group (15% vs. 0% and 2.5% vs. 0%, respectively). Troponin levels were more than 10 times higher than the upper limit of normal in 9.1% (9/99) of the cases, with this rate higher in the high-risk group than in the low-risk group (17.5% vs. 3.4%, respectively, *p* = 0.017). Myocarditis and pericardial effusion were also seen at higher rates in the high-risk group than in the low-risk group (*p* = 0.002 and *p* = 0.048, respectively). In the low-risk group, myocarditis and pericarditis were seen in 5.1% (3/59) and 6.8% (4/59) of the cases, respectively. Those cases were mild and their findings resolved rapidly during MIS-C treatment with no need of additional intervention. In the high-risk group, pericardial effusion, left ventricular dysfunction, and myocarditis was seen in 20% (8/40), 25% (10/40), and 27,5% (11/40) of the cases, respectively. Vasoactive and inotropic drugs were required in 37.5% (15/40) of high-risk cases. While coronary artery involvement (dilatation or aneurysm) was 10.1% (10/99) in the study group, coronary artery aneurysm was found only in 2% (2/99) of the patients. Sinus bradycardia was detected in 62.6% (62/99) of the cases during hospitalization. Sinus bradycardia appeared median two days after the fever subsided, and all cases resolved spontaneously within days without any intervention.

### 3.4. Treatments of Patients with MIS-C

The primary treatment regime was intravenous immunoglobulin (IVIG) plus methylprednisolone in 85.9% of patients. Nine patients received only IVIG, three received only steroids, and two did not receive IVIG or steroids. Additionally, six patients received anakinra and one patient received tocilizumab. Primary and supportive medical treatments were evaluated in Appendix A. When primary treatment regimens were compared in the risk groups (Table 4), it was seen that, more commonly, IVIG plus steroid treatments were administered in the high-risk group (*p* = 0.045).

### 3.5. Evaluation of the Performance of Laboratory Tests in Distinguishing Cases According to Risk Groups

Advanced statistical analyses were performed to determine the risk status of the cases according to the initial laboratory values. The variables that had a significant relationship with the univariate logistic regression model at the *p* < 0.10 level and did not correlate with each other were ALC (*p* = 0.055), ferritin (*p* = 0.034), BNP (*p* < 0.001), D-dimer (*p* = 0.080), total protein (*p* = 0.001), age (*p* = 0.036), and troponin (*p* = 0.017). These variables, which were found to be significant with the univariate logistic regression model, were included in the backward multiple logistic regression model. The model results are shown in Table 5. Based on the model, we found:An increase of 1000 units in BNP values increases the risk of being high-risk by 12.5%.A one-unit increase in total protein values reduces the risk of being high-risk by 36.8%.Those with a troponin value increased by ≥10-fold have a 9.5 times higher risk of being high-risk than those with a lower value.

We also evaluated several laboratory parameters’ univariate and combined test results in determining the risk groups. Combined methods show how accurately seven selected laboratory parameters classify risk groups together (Table 6). In other words, when a new patient arrives, the rate of assigning the person to the correct class based on these values is 85.9%, according to SVM. AUC values also show how well each biomarker performs the correct classification on its own. When the laboratory parameters were examined separately, BNP (AUC = 0.772), total protein (AUC = 0.750), and ferritin (*p* = 0.742) gave the best discrimination performance. The performances of seven laboratory parameters to discriminate the risk group together were also examined with different discrimination analysis methods. The best accuracy rate was obtained with the SVM and Naïve Bayes methods.

The contribution (importance) of biomarkers in differentiating risk groups with the SVM method is shown in Figure 4. The parameters that contributed the most in demonstrating the risk group were BNP, total protein, ferritin, D-dimer, ALC, and procalcitonin, respectively. It was observed that the number of platelets did not contribute.

## 4. Discussion

Two waves of cases were observed approximately one year apart in the study period. Notably, the total and high-risk cases decreased in the second wave (4Q of 2021) compared to the previous (4Q of 2020). In the USA, the second peak of MIS-C cases (September–October 2021) was lower than the previous one (December 2020–February 2021) [9]. In a preprint from southeast England, compared with the Alpha wave, they found fewer MIS-C cases relative to SARS-CoV-2 infections during both the pre-vaccine Delta, post-vaccine Delta, and Omicron waves [10]. The relative decrease in the incidence of MIS-C with the progression of the pandemic can be attributed to the characteristics of the new variants or the decrease in the number of uninfected children in the community. In the second peak period of 2021, although the Delta variant was thought to be dominant in our country, we could not assess this issue due to the lack of routine analysis of variants and the lack of sufficient data on seropositivity rates in the population.

Most MIS-C patients lacked a history of a previous SARS-CoV-2 infection or contact with a proven COVID-19 case. This situation is compatible with the literature [11,12]. In a pandemic, identifying exposure can be difficult as infected contacts may be asymptomatic or may never have been tested. Population seroconversion rates increase as more people become infected or vaccinated. As a result, the contribution of serological tests will gradually decrease in cases with a differential diagnosis of MIS-C [11]. The uncertainty of the epidemiological relationship is a handicap for the clinician in the differential diagnosis of MIS-C. This uncertainty can lead to overdiagnosis and misdiagnosis of MIS-C, leading to serious medical errors [13].

We found that the risk of complications (in terms of high-risk) was higher in cases aged 6 years and older. Although there was a male predominance in the cases, no effect of gender was observed in terms of the course of the disease. Studies identify older ages (especially the 6–12 years old group) as a risk factor for severity in MIS-C [11,12,14,15]. Male predominance is observed in MIS-C, but no effect of gender on prognosis is shown in systematic reviews [15,16,17]. Our results were compatible with the literature.

Abrams et al. found that ICU admission was more likely for patients who needed respiratory support at admission [15]. In two systematic reviews, the cumulative rates of chest imaging abnormalities were 13.7% and 35.5% [16,17]. In our study, an abnormal chest radiograph at admission was more common in the high-risk group (47.5%). Chest imaging, in our opinion, should be included in the first-step evaluation of MIS-C cases.

KD and MIS-C findings are similar in many respects. Although KD-like symptoms are associated with coronary involvement in MIS-C, they usually have a mild clinical course [6,16,17,18,19]. In our study, KD-like symptoms and other clinical and demographic features did not differ between risk groups.

Cardiac involvement in MIS-C is common and is one of the most important factors affecting prognosis [19,20,21]. Myocarditis, LV dysfunction, pericarditis, valvulitis, and coronary artery dilatation are the main cardiac complications. LV dysfunction, shock, and coronary artery involvement are reported in 31–64%, 50–80%, and 14–48% of MIS-C cases, respectively [11,12,17,18,22,23]. Alkan et al. reported valve insufficiencies in all clinical severity types of MIS-C, while ventricular dysfunction and coronary dilatation were seen in only severe cases [21]. Although the frequency of valvulitis was not different in the risk groups in our study, tricuspid and aortic valve involvements were more common in the high-risk group. In a review of short-term cardiovascular complications of MIS-C, the incidence of any valvulitis was reported as 24–48% [24]. In the same report, tricuspid and aortic valvulitis rates were 3.8–33% and 5.5%, respectively. As expected, myocarditis and pericarditis were seen more frequently in the high-risk group. On the other hand, we saw that isolated mild valve involvement might also occur in the low-risk group. Those findings may improve with MIS-C therapy without additional cardiac treatment.

Considering the general clinical course in MIS-C cases, the disease severity and ICU admission rates vary between studies. The incidence of ICU admission and mortality rate in MIS-C were reported as 31–80% and 0–2.5%, respectively [11,14,16,17,19,22,25]. In a recent multicentric report from India, Nayak et al. reported that ICU admission and mortality rates were 55.22% and 11.2%, respectively [12]. As of March 2022, the mortality rate in the USA reported by CDC was 0.8% [9]. In our study, the ICU admission rate was lower than average (15.2%), but this rate increased to 37.5% in the high-risk group. Approximately 40% of our cases had at least one complication or severe clinical course during their hospitalization. In their systematic review, Hoste et al. reported that 86% of the cases experienced severe course [16]. While the overall mortality was very low, it was observed that the presence of an uncontrolled pre-existing medical condition increased mortality in our study by seven-fold. In the Indian cohort, a pre-existing medical condition in discharged and exitus groups was 19.7% and 46.7%, respectively [12]. It is crucial to manage patients with co-morbidity with more intensive and aggressive treatment.

The conservative approach for mild cases of MIS-C is now being discussed more. However, a few clinical studies have compared treatments with and without IVIG [26,27,28]. Evaluating different treatment approaches is not the subject of this study. However, it is obvious that clinical features and laboratory parameters are needed to distinguish between cases with mild and severe clinical courses at the time of application for treatment optimization. We defined high- and low-risk groups by considering multiple factors. All clinically adverse outcomes were included in the high-risk group to increase the sensitivity in selecting high-risk cases, which was a priority for our study. We wanted to distinguish low-risk cases that could potentially be managed more conservatively (e.g., outpatient follow-up or medical treatment without IVIG). In our study, we used several modeling strategies, i.e., the univariate, multivariate, and machine learning approaches, in the classification task of risk groups. In the univariate analysis, ALC, BNP, ferritin, D-dimer, total protein, and troponin levels were different between risk groups. In the multivariate analysis, elevated BNP and troponin and low total protein levels were independent risk factors for high-risk. The univariate approaches were frequently used due to their simplicity and lowered complexity. However, the results might be biased in the presence of confounding factors. Moreover, it might be reasonable to combine several predictors (or biomarkers) and adjust for confounding factors to achieve higher predictive accuracy. In this study, when the performance of seven laboratory parameters in differentiating the risk group was evaluated, BNP, total protein, ferritin, and D-dimer showed the highest performance according to the ROC analysis results. We combined the laboratory tests using several machine learning models, e.g., CART, SVM with radial basis function, Naïve Bayes, and LDA, which were selected among others due to their flexibility, strong mathematical background, and popularity in the literature. Combined test results yielded higher classification accuracies compared to the single test results. In practice, it may not be possible to find one machine learning algorithm that performs the best in all performance measures. In such cases, one may select from available models by considering several performance measures. In this study, SVM and Naïve Bayes algorithms were the best in overall accuracy, while the logistic regression model resulted in the most sensitive results. In clinical studies and systemical reviews, the main biomarkers that are related to severe condition are elevated CRP [15,16], ferritin [11,15,16,21,25,29,30], BNP [15,21], D-dimer [15,16,31], troponin [15,16,21] levels, and reduced platelet [12] and lymphocyte counts [15,16,21]. Although it has been shown that various advanced markers such as IL-6, IL-10, IL-17A, and interferon-gamma (IFN-γ) have prognostic value, limited availability is the main obstacle to their widespread use [15,16,32].

The most important limitation of our study was the retrospective nature of the analysis. Our knowledge and experience about MIS-C increased day by day during the pandemic. In the early stages of the epidemic, the treatment of cases may have been more aggressive, or sometimes treatments may have been delayed. However, in our study, it is seen that a higher rate of IVIG and steroid combination therapy was administered to the high-risk group. Therefore, it is not possible to say that the choice of treatment causes a poor prognosis. Variants may be one of the critical confounding factors considering MIS-C incidence and clinical differences in the 2020 and 2021 waves. Evaluations made at the beginning of the epidemic or today may not be as valid for new MIS-C cases in the coming period. Different results in MIS-C studies may be related to differences in the study plan, time frame investigated, and population. Over time, how the differences in the course of COVID-19 in the population will reflect on the MIS-C clinic should be closely monitored. In our study, the levels of many different cytokines such as IL-6, IL-10, and interferon-gamma (IFN-γ) were not studied. Various biomarkers may perform much better. One of the primary purposes of conducting our study with the routine tests that many centers use was to increase the usability and reproducibility of the results in future studies. In relation to the last point, we only evaluated the laboratory markers at admission because we wanted to focus on the early identification of high-risk patients. Studies that examine temporal associations between laboratory markers and outcomes might help clarify these markers’ predictive potential.

## 5. Conclusions

In order to increase the cost effectiveness of MIS-C treatment protocols and create opportunities for more conservative approaches, it is necessary to distinguish low-risk patients who do not have a severe course and are not accompanied by complications. Our study showed that BNP, total protein, ferritin, and D-dimer laboratory tests had the highest performance in differentiating risk groups. Further prospective and comprehensive studies with larger cohorts are needed to identify low-risk cases to optimize MIS-C treatment guidelines.

## Figures and Tables

**Figure 1 jcm-11-04615-f001:**
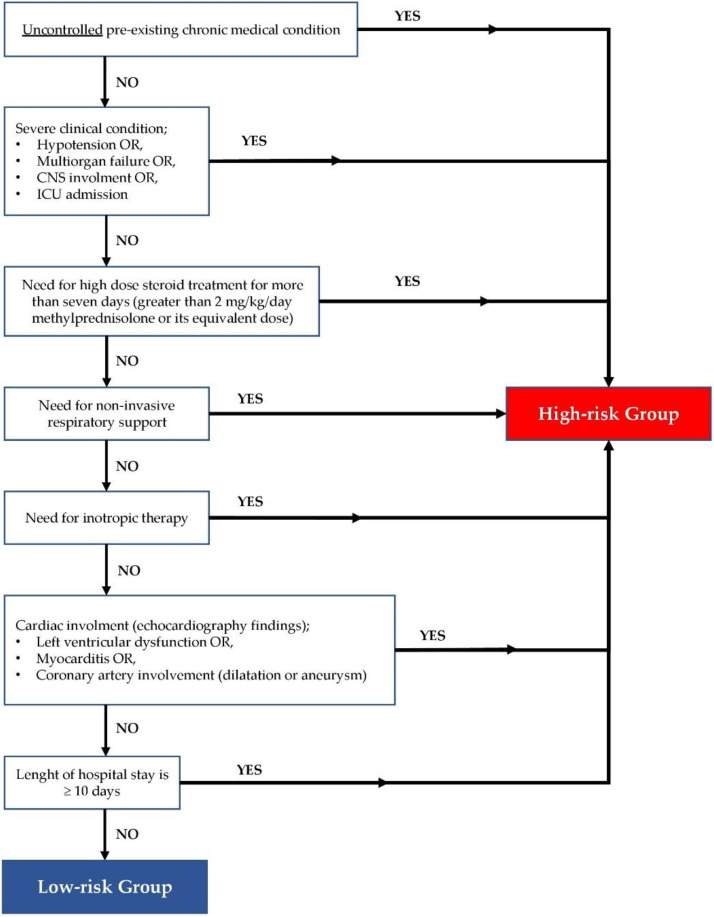
Flow diagram of the risk classification of MIS-C patients.

**Figure 2 jcm-11-04615-f002:**
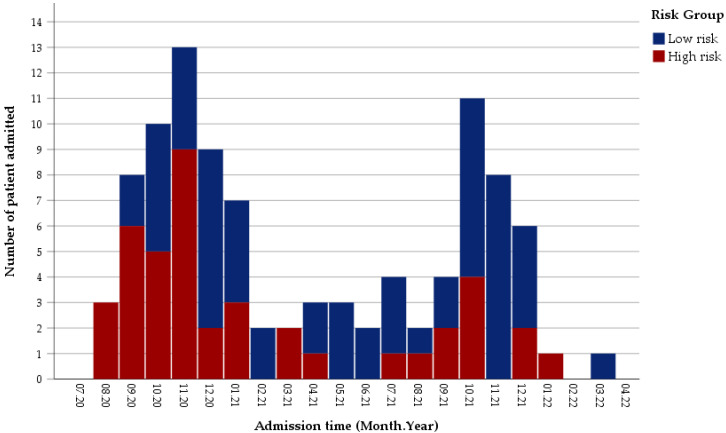
Distribution of patients in the study period.

**Figure 3 jcm-11-04615-f003:**
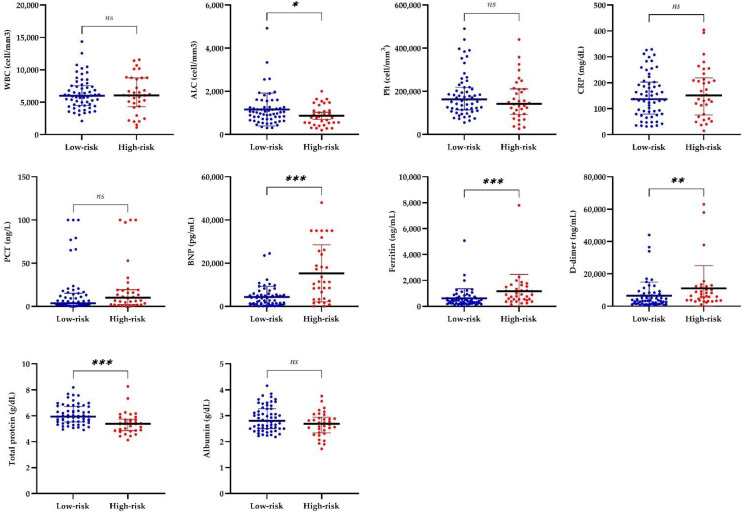
Comparison of laboratory values of the risk groups. Bars show mean ± SD if data were normally distributed (WBC and CRP) and median ± IQR if not (ALC, Plt, PCT, BNP, ferritin, D-dimer, total protein, and albumin). * *p* < 0.05, ** *p* < 0.005, *** *p* < 0.001. Abbreviations: ALC, absolute lymphocyte count; BNP, brain natriuretic peptide; CRP, C-reactive protein; ns, not significant; PCT, procalcitonin; Plt, platelet; WBC, white blood cell.

**Figure 4 jcm-11-04615-f004:**
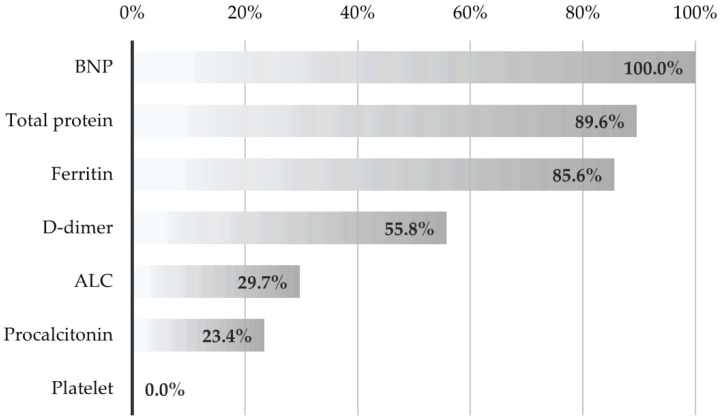
Contribution levels of selected laboratory values to the differentiating of the risk groups. Abbreviations: ALC, absolute lymphocyte count; BNP, brain natriuretic peptide.

**Table 1 jcm-11-04615-t001:** Characteristics and COVID-19-related data of MIS-C patients by risk groups.

	All MIS-C	Low-Risk Group	High-Risk Group	*p **
Number of patients, *n* (%)	99 (100)	59 (59.6)	40 (40.4)	
Age, month, median (min.–max.)	83 (3–205)	73 (16–191)	95.5 (3–205)	0.07
Age groups, *n* (%)				**0.03**
<72-month-old	37 (37.4)	27 (45.8)	10 (25.0)	
≥72-month-old	62 (62.6)	32 (54.2)	30 (75.0)	
Gender male, *n* (%)	63 (63.6)	38 (64.4)	25 (62.5)	0.84
Pre-existing medical condition, *n* (%)	10 (10.1)	1 (1.7)	9 (22.5)	N/A **
Autoinflammatory diseases	2 (2.0)	-	2 (5.0)	
Asthma	2 (2.0)	1 (1.7)	1 (2.5)	
Metabolic diseases	2 (2.0)	-	2 (5.0)	
Malignancy	2 (2.0)	-	2 (5.0)	
Congenital heart disease	1 (1.0)	-	1 (2.5)	
Chronic kidney disease	1 (1.0)	-	1 (2.5)	
Kawasaki Disease symptoms, *n* (%)				
Fever	99 (100)	59 (100)	40 (100)	-
Conjunctivitis	69 (69.7)	41 (69.5)	28 (70.0)	0.97
Rash	64 (64.6)	41 (69.5)	23 (57.5)	0.22
Oral mucous membrane changes	48 (48.5)	27 (45.8)	21 (52.5)	0.51
Peripheral extremity changes	30 (30.3)	14 (23.7)	16 (40.0)	0.08
Cervical lymphadenopathy (>1.5 cm)	13 (13.1)	8 (13.6)	5 (12.5)	0.87
Number of positive criteria (instead of fever) for Kawasaki Disease, *n* (%)				0.53
≤1	32 (32.3)	20 (33.9)	12 (30.0)	
2–3	52 (52.5)	32 (54.2)	20 (50.0)	
≥4	15 (15.2)	7 (11.9)	8 (20.0)	
Other symptoms and findings, *n* (%)				
Abdominal pain ***	59 (67.8)	36 (70.6)	23 (63.9)	0.51
Nausea/vomiting	64 (64.6)	38 (64.4)	26 (65.0)	0.95
Diarrhea	47 (47.5)	29 (49.2)	18 (45.0)	0.68
Headache ***	22 (30.6)	15 (37.5)	7 (21.9)	0.15
Respiratory distress	15 (15.2)	0 (0)	15 (37.5)	N/A **
Arthralgia/arthritis	14 (14.1)	9 (15.3)	5 (12.5)	0.70
Organomegaly (hepatomegaly or splenomegaly)	13 (13.1)	5 (8.5)	8 (20.0)	0.09
Neurological symptoms/findings (except headache)	7 (6.1)	0 (0)	7 (17.5)	N/A **
COVID-19 history, *n* (%)				0.97
Previous SARS-CoV-2 infection ^§^	25 (25.3)	15 (25.4)	10 (25.0)	
Close contact with a case but no infection	43 (43.4)	26 (44.1)	17 (42.5)	
No close contact or a history of infection	31 (31.3)	18 (30.5)	13 (32.5)	
The severity of previous SARS-CoV-2 infection ^†^, *n* (%)				0.15
Non-severe (no need for hospitalization)	23 (92.0)	15 (100)	8 (80.0)	
Need of hospitalization	2 (8.0)	0 (0)	2 (20.0)	
Duration between COVID-19 (or contact) and MIS-C ^‡^ week, median (min.–max.)	4 (2–16)	4 (2–16)	4 (2–9)	0.34
Duration between the onset of the MIS-C symptoms and admission, day median (min.–max.)	5 (1–10)	5 (1–10)	5.5 (1–10)	0.39
SARS-CoV-2 PCR and serology results at the admission, *n* (%)				0.23
PCR and serology are both positive	4 (4.0)	1 (1.7)	3 (7.5)	
PCR positive—serology negative	1 (1.0)	0	1 (2.5)	
PCR negative—serology positive	93 (93.9)	57 (96.6)	36 (90.0)	
PCR and serology are both negative	1 (1.0)	1 (1.7)	0	
Chest X-ray abnormal findings, *n* (%)	24 (24.2)	5 (8.5)	19 (47.5)	**<0.001**
Low blood pressure (hypotension/shock), *n* (%)	13 (13.1)	0 (0)	13 (32.5)	N/A **
Respiratory support (high flow oxygen or invasive mechanical ventilation) need, *n* (%)	15 (15.2)	0 (0)	15 (37.5)	N/A **
ICU admission, *n* (%)	15 (15.2)	0 (0)	15 (37.5)	N/A **
Length of stay, day median (min.–max.)	7 (3–24)	6 (3–9)	10,5 (6–24)	N/A **
Mortality, ratio (%)				
General mortality	3/99 (3.0)	0 (0)	3/40 (7.5)	**0.03**
Mortality with a pre-existing medical condition	2/10 (20)		2/9 (22.2)	
Mortality in previously healthy children	1/89 (1.1)		1/31 (3.2)	

* Significant values are shown in bold; ** N/A: Not applicable (These variables are accepted as components of the high-risk group); *** Abdominal pain and headache could not be evaluated in infants and accepted as a missing value. ^§^ SARS-Cov-2 infection confirmed with a PCR analysis which was performed during primary infection; ^†^ Analysis of 25 patients who had a history of COVID-19 confirmed by a PCR analysis; ^‡^ Analysis of 68 patients who had a history of infection or close contact with COVID-19. Abbreviations: COVID-19, Coronavirus disease 2019; ICU, intensive care unit; PCR, polymerase chain reaction.

**Table 2 jcm-11-04615-t002:** Laboratory findings of patients at admission.

Variables *	Patients with MIS-C ** (*n* = 93)	Reference Ranges
WBC (cell/mm^3^)	6281 ± 2592	
Absolute lymphocyte count (cell/mm^3^)	930 (200–4920)	
Absolute neutrophil count (cell/mm^3^)	4365 ± 2255	
Platelet count (cell × 10^3^/mm^3^)	152 (27–490)	
ESR (mm/h)	42.6 ± 24.4	0–20
C-reactive protein (mg/dL)	155.5 ± 90.0	0–0.5
Procalcitonin (ng/mL)	6.0 (0.1–100)	0–0.5
Ferritin (ng/mL)	534 (108–7800)	14–124
BNP (pg/mL)	4611 (52–48050)	0–125
LDH (U/L)	343 (61–4285)	120–300
ALT (U/L)	39 (7–838)	0–33
AST (U/L)	45 (9–965)	0–32
Total bilirubin (mg/dL)	0.41 (0.1–8.37)	0–0.9
Direct bilirubin (mg/dL)	0.20 (0.06–7.06)	0–0.3
Total protein (g/dL)	5.84 ± 0.84	6–8
Albumin (g/dL)	2.80 ± 0.49	3.2–4.5
Sodium (mEq/L)	131.7 ± 3.7	136–145
BUN (mg/dL)	13.4 (5.6–87.9)	5–18
Creatinine (mg/dL)	0.52 (0.24–3.19)	0.4–0.9
aPTT (s)	31.1 (22.3–58.0)	25–36
PT (s)	14.1 (11.4–62.3)	10–14
INR	1.26 (0.96–5.85)	0.8–1.2
D-dimer (ng/mL)	4620 (600–63000)	0–500
Fibrinogen (mg/dL)	486.9 ± 173.4	180–350

* Values were reported as mean ± standard deviation if data were normally distributed; if not, they were presented as median with minimum and maximum values. ** Six cases with chronic diseases (two malignancies, two autoinflammatory, one chronic kidney disease, and one metabolic disease) who have baseline abnormalities before MIS-C were excluded during the analysis of laboratory values. Abbreviations: ALT, alanine aminotransferase; AST, aspartate aminotransferase; aPTT, activated partial thromboplastin time; PT, prothrombin time; BNP, brain natriuretic peptide; BUN, blood urea nitrogen; ESR, erythrocyte sedimentation rate; INR, international normalized ratio; LDH: lactate dehydrogenase; WBC, white blood cell.

**Table 3 jcm-11-04615-t003:** Cardiac findings of patients with MIS-C.

Findings *n (%) **	All MIS-C (*n* = 99)	Low-Risk MIS-C (*n* = 59)	High-Risk MIS-C (*n* = 40)	*p ***
Valvulitis	72 (72.7)	41 (69.49)	31 (77.5)	0.38
Valve involved Mitral Tricuspid Aortic Pulmonary	71 (71.7) 25 (25.3) 10 (10.1) 3 (3.0)	41 (69.5) 8 (13.6) 2 (3.4) 0 (0)	30 (75.0) 17 (42.5) 8 (20.0) 3 (7.1)	0.55 **0.001** **0.007** 0.06
Number of valves involved None 1 2 >2	27 (27.3) 41 (41.4) 24 (24.2) 7 (7.1)	18 (30.5) 30 (50.8) 11 (18.6) 0 (0)	9 (22.5) 11 (27.5) 13 (32.5) 7 (17.5)	**0.004**
Valvular maximum insufficiency degree None 1 2 3	27 (27.3) 65 (65.7) 6 (6.1) 1 (1.0)	18 (30.5) 41 (69.5) 0 (0) 0 (0)	9 (22.5) 24 (60.0) 6 (15.0) 1 (2.5)	**0.011**
Left ventricular dysfunction	10 (10.1)	0 (0)	10 (25.0)	N/A
Troponin levels Normal or <10-fold of the upper limit Increased by ≥10-fold of upper limit	90 (90.9) 9 (9.1)	57 (96.6) 2 (3.4)	33 (82.5) 7 (17.5)	**0.017**
Myocarditis	14 (14.1)	3 (5.1)	11 (27.5)	**0.002**
Pericardial effusion	12 (12.1)	4 (6.8)	8 (20.0)	**0.048**
Coronary artery involvement	10 (10.1)	0 (0)	10 (25.0)	N/A
Transient sinus bradycardia	62 (62.6)	34 (57.6)	28 (70.0)	0.21
Duration between fever ending and bradycardia starting, median day (min.–max.)	2 (0–6)	1.5 (0–3)	2 (0–6)	0.15

* Percentage values are given within their risk group. ** Significant values are shown in bold. N/A: Not applicable (these variables were identified as the distinguishing feature in defining the groups).

**Table 4 jcm-11-04615-t004:** Primary treatment regimens of patients within MIS-C risk subgroups.

Treatment Regime, *n (%)*	Low-Risk (*n* = 59)	High-Risk (*n* = 40)	*p*
IVIG plus steroid	46 (78.0)	39 (97.5)	0.045 *
IVIG only	8 (13.6)	1 (2.5)	
Steroid only	3 (5.1)	0 (0)	
IVIG or steroid not used	2 (3.4)	0 (0)	

* Chi-square test with Monte Carlo method was used. Abbreviation: IVIG, intravenous immunoglobulin.

**Table 5 jcm-11-04615-t005:** Results of multiple logistic regression model.

Variable	OR	95% CI for OR	*p **
BNP (10^−3^)	1.125	1.047–1.209	**0.001**
Total protein **	2.717	1.130–6.536	**0.026**
Troponin level (Increased by ≥10-fold upper normal limit)	9.491	1.287–70.018	**0.027**

* Significant values are shown in bold; ** Small values show a higher risk. Abbreviation: BNP, brain natriuretic peptide.

**Table 6 jcm-11-04615-t006:** Univariate and combined test results of several laboratory parameters in the classification of the risk groups.

Method	Cut-Off Value	Accuracy	Sensitivity	Specificity	PPV	NPV	AUC
Univariate ROC Analysis							
Procalcitonin	4.5	0.607	0.706	0.509	0.453	0.75	0.614
Platelet	73,000	0.429	0.824	0.034	0.329	0.25	0.566
Ferritin	546	0.721	0.765	0.678	0.578	0.833	0.742
BNP	8332	0.77	0.676	0.864	0.742	0.823	0.772
ALC	780	0.606	0.5	0.712	0.5	0.712	0.627
D-dimer	2890	0.674	0.912	0.407	0.478	0.923	0.681
Total protein	5.48	0.69	0.618	0.763	0.6	0.776	0.75
**Combined Tests Analysis (Machine Learning)**							
Decision Tree (CART)		0.77	0.676	0.864	0.742	0.823	
SVM w/radial basis function		0.859	0.735	0.983	0.962	0.866	
Logistic regression		0.785	0.823	0.746	0.651	0.88	
Naïve Bayes		0.836	0.706	0.966	0.923	0.851	
LDA		0.746	0.559	0.932	0.826	0.786	

Abbreviations: ALC, absolute lymphocyte count; AUC, area under the curve; BNP, brain natriuretic peptide; CART: classification and regression tree; LDA, linear discriminant analysis; NPV, negative predictive value; PPV, positive predictive value; SVM, support vector machine.

## Data Availability

The data presented in this study are available on request from the corresponding author.

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
