# Peer review of "Evaluation of Baseline Characteristics and Prognostic Factors in Multisystemic Inflammatory Syndrome in Children: Is It Possible to Foresee the Prognosis in the First Step?"

_jcm, 2022, doi:10.3390/jcm11154615_

Round 1
Reviewer 1 Report
This was a well written paper that describes the experience in Erciyes University, but is reflected in other Western Nations. In this retrospective study of 99 patients with MIS-C hospitalized between August 2020 and March 2022, their goal was to prognosticate the clinical severity to aid in outcome prediction and determine the therapy necessary. To do this, they divided the patients into two groups, those with high-risk and low-risk based on clinical and laboratory presentation. They concluded that “BNP, total protein, and troponin were associated with higher risk, but when the laboratory parameters were used together, BNP, total 38 protein, ferritin, and D-dimer provided the highest contribution to the discrimination of the risk groups (100%, 89.6%, 85.6%, and 55.8%, respectively)”.
This is study is an important one since it is vital to determine which patients can be safely monitored and which must be treated urgently to prevent complications. I agree with their conclusion that more modeling studies with larger sample size is needed, but this is a good start. It mirrors the experience in the experience at our center and provides guidelines for those who are seeing the patient initially to most appropriately direct their care.
The figures were well done and easy to interpret. The tables could be presented more clearly and, although the normal values in all labs are similar, it would be helpful to include the normal values in their laboratory. It would have been most interesting to see the cytokine levels and perhaps this could be included in future studies. This might add another parameter to clarify the risk groups.
The cardiac findings were of interest and follow-up would be of interest. There is a discussion of rapid resolution in the low-risk groups, but further discussion on the high risk groups would be of interest.
The statistics appear to be well done and valid, but review with a statistician would be helpful. More discussion of the machine based learning and comparison with the more conventional statistics.
Author Response
Response to Reviewer 1 Comments
We would like to thank the Reviewers for taking the necessary time and effort to review the manuscript. We sincerely appreciate all your valuable comments and suggestions, which helped us improve the manuscript's quality.
Point 1: The figures were well done and easy to interpret. The tables could be presented more clearly, and although the normal values in all labs are similar, it would be helpful to include the normal values in their laboratory.
Response 1: We revised table 2 and added reference ranges
Point 2: It would have been most interesting to see the cytokine levels and perhaps this could be included in future studies. This might add another parameter to clarify the risk groups.
Response 2: As the reviewer pointed out, we revised our limitations and added; “In our study, the levels of many different cytokines such as IL-6, IL-10, and interferon-gamma (IFN-γ) were not studied. Various biomarkers may perform much better. One of the primary purposes of conducting our study with the routine tests that many centers use was to increase the usability and reproducibility of the results in future studies.” (line 442)
Point 3: The cardiac findings were of interest, and follow-up would be of interest. There is a discussion of rapid resolution in the low-risk groups, but further discussion on the high-risk groups would be of interest.
Response 3: In the cardiac findings section, we added, “In the high-risk group, pericardial effusion, left ventricular dysfunction, and myocarditis was seen in 20% (8/40), 25% (10/40), and 27,5% (11/40) of the cases, respectively. Vasoactive and inotropic drugs were required in 37.5% (15/40) of high-risk cases” (line 235). We did not evaluate the detailed clinical course of myocardial and pericardial involvements (maximal involvement time after admission, details of the healing process, responses to the treatments and long-term course, cardiac MRI findings, etc.).
Point 4: The statistics appear to be well done and valid, but a review with a statistician would be helpful. More discussion of machine-based learning and comparison with more conventional statistics.
Response 4: The methodology was detailed by a statistician who was also among the authors. The discussion about the differences between machine learning models and conventional statistics were revised.
Added parts can be seen on the following lines;
- Line 404 “..we used several modeling strategies, i.e., the univariate, multivariate, and machine learning approaches, in the classification task of risk groups”
- Line 409 “The univariate approaches were frequently used due to its simplicity and lowered complexity. However, the results might be biased in presence of confounding factors. Moreover, it might be reasonable to combine several predictors (or biomarkers) and adjust for confounding factors to achieve higher predictive accuracy”
- Line 415 “We combined the laboratory tests using several machine learning models, e.g., CART, SVM with radial basis function, Naïve Bayes, and LDA which are selected among others due to flexibility, strong mathematical background, and popularity in the literature. Combined test results yielded higher classification accuracies as compared to the single test results. In practice, it may not be possible to find one machine learning algorithm that performs the best in all performance measures. In such cases, one may select among from available models by considering several performance measures. In this study, SVM and Naïve Bayes algorithms were the best in overall accuracy, while the logistic regression model resulted in the most sensitive results”
Reviewer 2 Report
Overall, I really liked the paper, and find it a useful addition to the literature on MIS-C.
I'm not sure the section on treatment is relevant as the goal of the study is to define low versus high risk patients on admission. The authors themselves, state: "Evaluating different treatment approaches is not the subject of this study." As the treatment protocols were decided arbitrarily, and the outcomes were not evaluated, I think this section should be omitted.
Otherwise, I support publishing the manuscript.
Lastly, there are two small typos on lines 367 and 411.
Author Response
Response to Reviewer 2 Comments
We would like to thank the Reviewers for taking the necessary time and effort to review the manuscript. We sincerely appreciate all your valuable comments and suggestions, which helped us improve the manuscript's quality.
Point 1: I'm not sure the section on treatment is relevant as the goal of the study is to define low versus high risk patients on admission. The authors themselves, state: "Evaluating different treatment approaches is not the subject of this study." As the treatment protocols were decided arbitrarily, and the outcomes were not evaluated, I think this section should be omitted.
Response 1: Although our study does not aim to compare treatment approaches, we think that the treatment given to the cases may be requested by the reader, just like demographic information. We simplified the treatment section in line with the reviewer's recommendations. As a picture of the primary treatment approach, it would be appropriate to include Table 4 as a supplement (Table S1). The other table (originally Table 5, in the revised version Table 4) shows the distribution of the primary treatment regimen among the groups. We think it would be appropriate to leave this table in the main text to indicate that an incomplete treatment was not given in high-risk patients compared to low-risk patients.
Point 2: There are two small typos on lines 367 and 411
Response 2: We reviewed and corrected typos.
Reviewer 3 Report
As the world continues through the global pandemic due to SARS-CoV2 our understanding of the disease processes continues to grow. Multisystem Inflammatory Disease in Children (MIS-C) is a well described phenomenon, but our understanding of its pathophysiology remains incomplete. Fortunately, while there is a significant degree of short-term morbidity, mortality remains low.
The authors describe use their experience in attempt to determine early which patients will have a more severe disease course. One of their well-stated reasons that this is important is that treatment can be modified in those patients at lower risk of more severe disease. Approximately 40% of their patients had a complication and understanding who is likely to do so is a worthy effort.
I think this information is a valuable addition to the literature, in part because their experience is not identical to that of others. In particular their patients rate of cardiac ventricular dysfunction was only 10% this is typically reported as greater than 50%. Their population experienced a mortality rate of 3.2% which is slightly higher than the typically reported 1-2%; in the more ill group the mortality rate was 7.5%.
Overall, their data seem to indicate that a greater degree of inflammation at presentation as demonstrated by high serum levels of d-dimer, ferritin and natriuretic peptide predict a more severe clinical course. Why these and not other markers of inflammation is not clear.
There are several aspects of how this data is presented that could be improved to make the authors' findings clear.
A) they refer to the groups as "high risk" and "low risk". What they mean is less or greater illness and they are evaluating how to identify those at higher risk to have more severe illness.
B) on page 2, Materials and Methods, I believe they mean to indicate their criteria for "severe clinical condition". I believe this is the more ill clinical course of MIS-C, yet on line 67 they state "at follow-up". Were some of their patients still this ill at follow-up? Figure 1 makes their division into classification much easier to understand.
C) definition of "severe clinical condition" includes hypotension, multi organ failure, CNS involvement (other than isolated cephalgia) or ICU admission. Forty percent of their patients met this definition but on 15% based on ICU admission. As these criteria can overlap, I cannot which of their criteria was most common in patients in the more ill group.
D) page 11, line264 the authors chose a statistically significant cutoff of 0.10, why not the more typical 0.05?
Author Response
Response to Reviewer 3 Comments
We would like to thank the Reviewers for taking the necessary time and effort to review the manuscript. We sincerely appreciate all your valuable comments and suggestions, which helped us improve the manuscript's quality.
Point 1: Overall, their data seem to indicate that a greater degree of inflammation at presentation as demonstrated by high serum levels of d-dimer, ferritin and natriuretic peptide predict a more severe clinical course. Why these and not other markers of inflammation is not clear.
Response 1: We stated in the study limitations that we could not look at some biomarkers in the literature due to the limited facilities of the center (line 442). Although the possible reasons for this difference in biomarkers are an important point of discussion, we did not address this issue to avoid distracting the main focus of our study.
Point 2: They refer to the groups as "high risk" and "low risk". What they mean is less or greater illness and they are evaluating how to identify those at higher risk to have more severe illness.
Response 2: To clarify this issue, we added “The feature that we considered as risk here was defined as the risk of a more severe course of the disease. “ in the materials and methods section (Line 67).
Point 3: On page 2, Materials and Methods, I believe they mean to indicate their criteria for "severe clinical condition". I believe this is the more ill clinical course of MIS-C, yet on line 67 they state "at follow-up". Were some of their patients still this ill at follow-up? Figure 1 makes their division into classification much easier to understand.
Response 3: As the reviewer pointed out, we replaced the word "follow-up" with "hospitalization" to avoid misunderstanding at this point. When classifying the cases, those considered "high-risk" were those with more severe disease, as shown in Figure 1 (Line 69, 241 and 388).
Point 4: Definition of "severe clinical condition" includes hypotension, multi organ failure, CNS involvement (other than isolated cephalgia) or ICU admission. Forty percent of their patients met this definition but on 15% based on ICU admission. As these criteria can overlap, I cannot which of their criteria was most common in patients in the more ill group.
Response 4: Our high-risk group cases consisted of those with severe clinical conditions and features such as the need for non-invasive respiratory support, high-dose steroid therapy, or prolonged hospitalization, as shown in Figure 1. Although the need for intensive care was 15.1%, the high-risk case rate had increased to 40.4% when the other criteria were considered. All these criteria overlapped mainly and are stated in Tables 1, 3, and S1 separately. We added multiple organ failure and prolonged hospitalization rates; “MOF rate was 9% (9/99)”and “Twenty-three cases (23.2%) had a hospital stay of 10 days or longer.” (Line 165 and 166).
In addition, according to the reviewer's comments, the most common clinical conditions that caused the cases to be included in the high-risk group were also added in the results section; “The most common clinical conditions that caused the cases to be included in the high-risk group were prolonged hospitalization (23/99), respiratory failure (15/99), and hypotension (13/99), respectively.” (Line 167).
Point 5: Page 11, line264 the authors chose a statistically significant cutoff of 0.10, why not the more typical 0.05?
Response 5: We detailed our statistically methods in “Statistical Analyses” section as; “Our aim was to explore significant predictors of risk groups. In the first step, we performed a simple logistic regression model for each predictor separately. The predictors with p-values below 0.10 were selected as possible risk factors and the set of all possible predictors was included in the multiple logistic regression model. In this model, we adjusted the effect of each predictor by the remaining set, and the statistical significance level was set as p<0.05 in the final model. To conclude, p<0.10 was used to find the list of predictors via the crude model while p<0.05 was used to find the significant predictors in the multiple models (i.e., covariate-adjusted model).” (New added part, line 105 – 113)